# Starvation promotes concerted modulation of appetitive olfactory behavior via parallel neuromodulatory circuits

Kang I Ko[1], Cory M Root[1], Scott A Lindsay[2], Orel A Zaninovich[1],
Andrew K Shepherd[1], Steven A Wasserman[2], Susy M Kim[1], Jing W Wang[1]*

[1]Neurobiology Section, Division of Biological Sciences, University of California, San Diego, La Jolla, United States; [2]Cell and Developmental Biology Section, Division of Biological Sciences, University of California, San Diego, La Jolla, United States

**Abstract** The internal state of an organism influences its perception of attractive or aversive stimuli and thus promotes adaptive behaviors that increase its likelihood of survival. The mechanisms underlying these perceptual shifts are critical to our understanding of how neural circuits support animal cognition and behavior. Starved flies exhibit enhanced sensitivity to attractive odors and reduced sensitivity to aversive odors. Here, we show that a functional remodeling of the olfactory map is mediated by two parallel neuromodulatory systems that act in opposing directions on olfactory attraction and aversion at the level of the first synapse. Short neuropeptide F sensitizes an antennal lobe glomerulus wired for attraction, while tachykinin (DTK) suppresses activity of a glomerulus wired for aversion. Thus we show parallel neuromodulatory systems functionally reconfigure early olfactory processing to optimize detection of nutrients at the risk of ignoring potentially toxic food resources.

*For correspondence: jw800@ucsd.edu

Competing interests: The authors declare that no competing interests exist.

## Introduction

Sensory systems undergo dramatic functional modifications when animals enter different internal states such as hunger or arousal (*Su and Wang, 2014*). These functional changes in neural circuits support flexible and adaptive behaviors in animals in response to their changing needs and circumstances. The mechanisms driving these neural circuit modifications are under intense investigation and are fundamental to our understanding of how neural circuits support animal cognition and behavior.

Neuromodulators are an important class of molecules positioned to exert significant influence over local neural circuits through their effects on neuronal excitability and network properties (*Bargmann and Marder, 2013*). The study of hunger and satiety in fruit flies has identified key neuromodulators that serve to communicate an animal's nutritional status to its nervous system and thus, change its behavior. Neuromodulators such as dopamine, short neuropeptide F (sNPF) and NPF have been demonstrated to influence different aspects of appetitive behaviors such as taste sensitivity (*Inagaki et al., 2012*; *Marella et al., 2012*; *Inagaki et al., 2014*), the formation and expression of appetitive memories (*Krashes et al., 2009*), odor preference (*Root et al., 2011*; *Beshel and Zhong, 2013*), and control of food intake (*Lee et al., 2004*; *Yu et al., 2004*; *Wu et al., 2005*; *Wang et al., 2013*).

In natural environments, foraging and feeding behaviors expose animals to risks of predation and harmful toxins in food. For example, food deprivation increases an animal's tolerance for noxious stimuli (*Gillette et al., 2000*; *Wu et al., 2005*; *Inagaki et al., 2014*) and suppresses escape behavior at the risk of predation (*Gillette et al., 2000*; *Gaudry and Kristan, 2009*). Thus, when evaluating potential food sources, animals must weigh both aversive and attractive sensory inputs. Their perceptions of these sensory stimuli and behavioral decisions are influenced by their own internal states and needs for

**eLife digest** Animals typically need to forage for their food, but doing so is not without risk. Foraging can expose an animal to predators and harmful toxins. Many animals use odors and other chemical signals to help them locate food or to avoid harm. In some animals, such as fruit flies, different parts of the nervous system are hardwired to encourage individuals to move towards attractive odors or away from unpleasant ones.

Fruit flies feed on the yeast that grows on decaying fruit. They do so by ignoring fresh fruits (which have very little yeast) and avoiding overly-rotten fruits (which might contain toxic chemicals). To determine ripeness, flies use a fruit's vinegar levels: fresh fruits contain low levels of vinegar, while fermented fruits have high levels. Previous studies using low levels of vinegar have shown that well-fed flies largely ignore the scent, while starving flies are attracted to it.

Ko et al. have built on the results of previous studies and now report that starving fruit flies are much less sensitive to unfavorable odors in high levels of vinegar and much more sensitive to favorable odors in low levels of vinegar. This behavior is due to two neuropeptides (molecules that carry signals between neurons) that have opposite effects on different parts of the fly's nervous system. One of the neuropeptides made the groups of neurons that respond to attractive odors more responsive, while the other suppressed the activity of neurons that normally respond to unpleasant odors. Together these changes could encourage the animals to take more risks when they are hungry, by suppressing of their ability to recognize noxious or harmful chemicals in favor of their ability to perceive attractive odors.

The effect of both neuropeptides is triggered by the insulin hormone, which carries information about the metabolic state (for example, whether it is starving or well-fed) throughout the whole animal. Thus, individual neurons may read the same metabolic signals and then respond in different ways to fine-tune the activity of nearby circuits of neurons to alter foraging behavior in a coordinated manner. Furthermore, it is almost certain that similar changes to the sensory system could affect an animal's appetite for food. One of the next challenges will be to attempt to understand if and how appetite in humans might be controlled in a similar way.

energy homeostasis. In *Drosophila*, starvation heightens sensitivity in odorant receptor neurons (ORNs) that are critical for behavioral attraction to appetitive odors through neuromodulatory mechanisms (*Root et al., 2011*). Starvation has also been shown to reduce behavioral avoidance to innately aversive odors (*Bracker et al., 2013*). This starvation effect requires the presence of appetitive odors and has been suggested to occur at higher order levels of the *Drosophila* brain (*Bracker et al., 2013*). Whether starvation also reduces sensitivity of ORNs in the periphery that are critical for behavioral avoidance is unknown.

Here we describe a neuromodulatory mechanism in *Drosophila* controlling the reduction of aversive odor sensitivity during starvation at the level of the first olfactory synapse. This pathway operates in parallel and independently of the mechanisms driving increases in attractive odor sensitivity. We show here that starvation does not simply scale up or down global activity in the antennal lobe. Rather, it upregulates activity in certain sensory channels and downregulates it in others in what appears to be an optimization strategy that may serve to increase the hedonic value of food odors. Thus, individual neurons may read the same global metabolic signals and differentially respond in a manner that fine tunes local circuits towards a concerted modulation of appetitive behaviors.

## Results

### The role of two neuropeptide modulatory systems in food search behavior

In *Drosophila*, some olfactory sensory channels are hardwired for innate behaviors (*Suh et al., 2004*; *Kurtovic et al., 2007*; *Semmelhack and Wang, 2009*; *Ai et al., 2010*; *Grosjean et al., 2011*; *Stensmyr et al., 2012*). The hedonic value of behaviorally relevant odors is therefore encoded by glomerular activity in the early olfactory system. In particular, activation of the DM1 glomerulus, innervated by Or42b ORNs, triggers attraction to an odor, while activation of DM5, innervated by

Or85a ORNs, reduces attraction to high concentrations of vinegar (*Semmelhack and Wang, 2009*). Thus, activity in DM1 and DM5 represents positive and negative valence, respectively, and physiological modulation of these glomeruli should alter olfactory behaviors.

To study how attractive and aversive odor input channels might be modulated in starved flies, we took advantage of the finding that the odor map changes as concentrations of odor increase (*Wang et al., 2003*). At intermediate concentrations, food odors such as vinegar are attractive to hungry flies. At low or high concentrations, however, odors are ignored (*Semmelhack and Wang, 2009*; *Root et al., 2011*). Using a single fly food odor search paradigm (*Root et al., 2011*; *Zaninovich et al., 2013*), we first sought to extend our earlier study which used low concentrations of vinegar (*Root et al., 2011*) and evaluated how behavioral attraction in starved flies changes in response to a broader range of odor concentrations from low to high.

We measured the time required for starved and fed flies to locate a source of apple cider vinegar across a range of concentrations. Starved flies typically locate the odor source within minutes after entering the observation chamber (*Figure 1A*). This behavior can be quantified by an appetitive index, which we define as the percentage of flies reaching the odor source within a 10 min observation period. In starved flies, the appetitive index rises (from 23 to 60) as the vinegar concentration increases (0.5–25%), but then steadily declines at higher concentrations (*Figure 1B*). At all tested concentrations (0.5–100%), the appetitive index in starved flies is greater than that of the fed flies.

sNPF, a *Drosophila* homolog of the mammalian orexigenic neuropeptide Y (NPY) (*Barsh and Schwartz, 2002*; *Lee et al., 2004*), enhances the olfactory sensitivity of ORNs and increases appetitive behavior at low odor concentrations (*Root et al., 2011*). To what extent does sNPF signaling account for behavioral attraction at all odor concentrations? To answer this question, we used RNAi to knock down expression of the sNPF receptor (sNPFR) in ORNs using *Orco-Gal4* (*Larsson et al., 2004*; *Vosshall and Hansson, 2011*). At low concentrations (0.5–1%) of cider vinegar, knockdown of sNPFR eliminated the behavioral difference between starved and fed flies (*Figure 1C*). However, at high concentrations (5–25%), the appetitive index of sNPFR knockdown flies remains significantly higher than that of the control fed flies. In a fed state, sNPFR knockdown had no behavioral effect compared to the control flies (*Figure 1—figure supplement 1A*). These results suggest the existence of a parallel mechanism for starvation-mediated changes in appetitive behavior at high odor concentrations.

One potential mechanism for regulating appetitive behavior at high odor concentrations is to alter a sensory neuron's neuropeptide response by modulating levels of a neuropeptide receptor (*Root et al., 2011*). Most neuromodulators signal through G-protein coupled receptors (GPCRs) (*Greengard, 2001*) and a small increase in GPCR expression can have a dramatic effect on neural circuit function and related behavior (*Bendesky et al., 2011*). We therefore performed a transcriptome analysis using RNA-seq to identify differentially expressed GPCRs in the antennae of fed and starved flies. We focused on the typical GPCR families (*Brody and Cravchik, 2000*) that include receptors for biogenic amines, neuropeptides, as well as classic neurotransmitters, because these types of GPCRs have the potential to alter excitability (*Greengard, 2001*).

This analysis identified 34 GPCRs that have higher expression in the antennae of starved flies (*Table 1*). This group included sNPFR, a GPCR we had previously shown to undergo upregulation in Or42b ORNs after starvation (*Root et al., 2011*). Other upregulated GPCRs, not yet described in ORNs, include the dopamine 2-like and dopamine-ecdysone receptors which have both been implicated in enhancing fly sugar receptor sensitivity in the starved state (*Inagaki et al., 2012*; *Marella et al., 2012*). Additional GPCRs shown to influence feeding behaviors include the serotonin 2A receptor (*Gasque et al., 2013*), a receptor that promotes anorectic behaviors and the GABA-B receptor type 1 (*Bjordal et al., 2014*), a receptor that is part of a circuit that detects amino acid imbalances. Whether these receptors are expressed in select glomeruli remains to be determined. Interestingly, dopaminergic (*Riemensperger et al., 2005*) and serotonergic (*Roy et al., 2007*; *Dacks et al., 2009*) terminals have been described in the adult antennal lobe, thus making these receptor classes highly plausible targets for internal state modulation.

We also identified 11 GPCRs that exhibited reduced expression in the antennae of starved flies (*Table 1*). This group included a few GPCRs described as having roles in feeding behaviors or energy storage. One example is as the adipokinetic hormone receptor, which regulates lipid and carbohydrate storage (*Bharucha et al., 2008*). Another is the pyrokinin 1 receptor, which is activated by hugin, a neuropeptide that suppresses feeding (*Melcher and Pankratz, 2005*). Although it is not known whether these two antennal GPCRs are expressed in ORNs, such localization has been

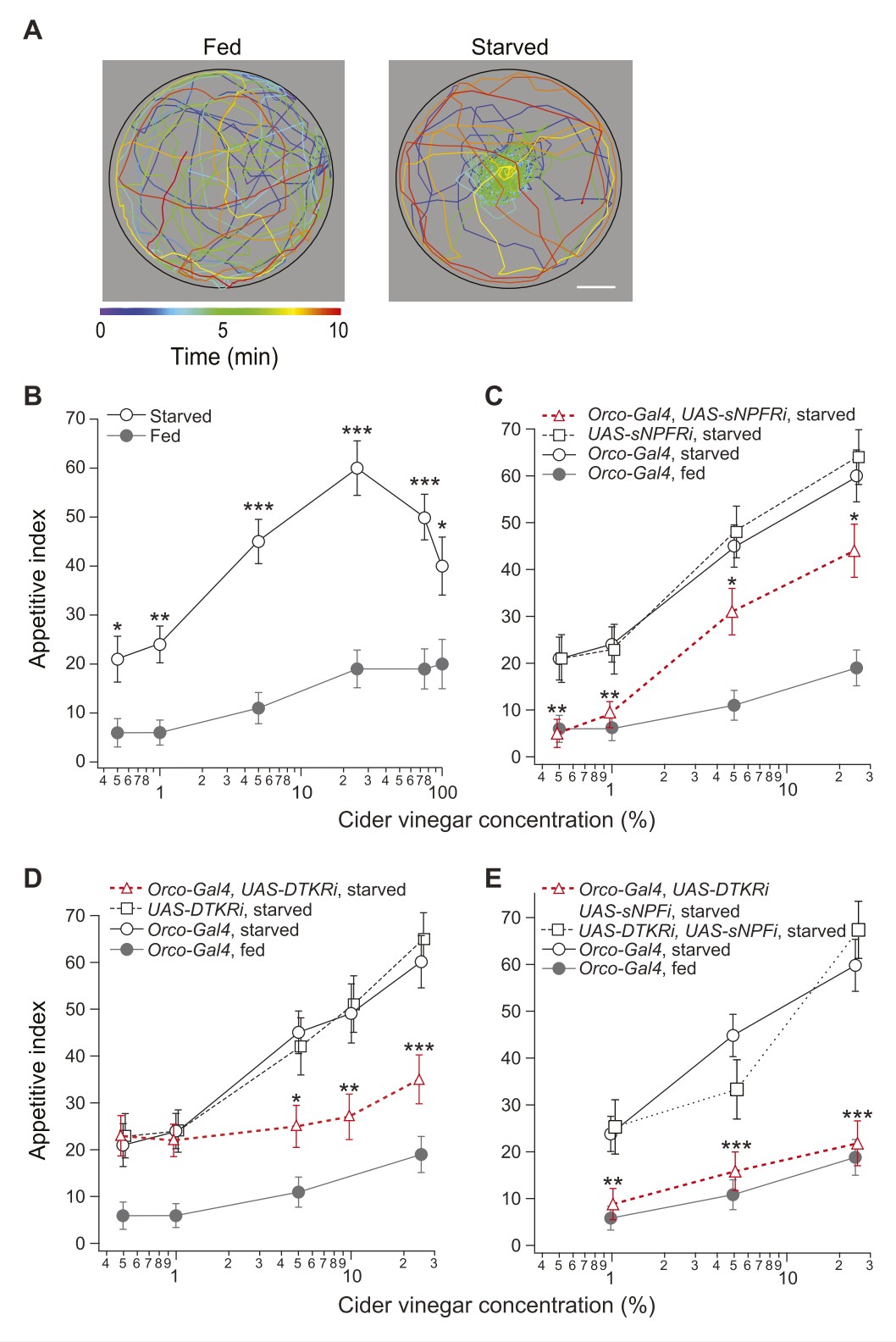

**Figure 1**. Starvation state fine-tunes appetitive behavior. (**A**) A single fly assay was used to measure food search behavior. The coordinates of representative fed (left) and starved (right) flies show their positions during a 10-min period in response to 5% cider vinegar. Scale bar: 10 mm. (**B**) The appetitive index of fed and starved *Orco-Gal4* control flies at varying concentrations. (**C–E**) The appetitive index of receptor knockdown flies, in which the receptor

*Figure 1. continued on next page*

*Figure 1. Continued*

RNAi is expressed in the *Orco* odorant receptor neurons (ORNs). (**C**) Short neuropeptide F receptor (sNPFR) knockdown flies. (**D**) *Drosophila* tachykinin receptor (DTKR) knockdown flies. (**E**) sNPF and DTKR dual knockdown flies. (**B**–**E**) n = 63–129 for each condition. Error bars show s.e.m. *p < 0.05, **p < 0.01, ***p < 0.001; z-test for proportions comparing between starved and fed (**B**), and comparing knockdown flies to the Orco-Gal4 and UAS-control group in the starvation state (**C**, **D**, **E**).

The following figure supplement is available for figure 1:

**Figure supplement 1**. Food search behavior in control and knockdown flies.

reported for the *Drosophila* tachykinin receptor (DTKR). This receptor, also known as Takr99D, mediates presynaptic inhibition (*Ignell et al., 2009*) and is implicated in nutritional stress responses (*Winther and Nassel, 2001*). We therefore focused our attention on this receptor.

To assay whether DTKR expression in ORNs is required for the post-fasting behavioral modification, we knocked down DTKR and measured the appetitive index. Starved DTKR knockdown flies do not behave differently from control starved flies in response to low concentrations (0.5–1%) of cider vinegar. In contrast, their appetitive index at high concentrations (5–25%) of cider vinegar is significantly lower than that of control flies (*Figure 1D*). Furthermore, we found no effect of DTKR knockdown on the behavior of fed flies (*Figure 1—figure supplement 1B*). Thus, DTKR signaling is necessary for starvation-dependent change in food search behavior at high, but not low odor concentrations.

Our results indicate both sNPFR and DTKR signaling contribute to appetitive changes. Do these two pathways fully account for the starvation response? To address this question, we explored whether removal of both signaling mechanisms transforms the behavior of starved flies into that of fed flies. Indeed, simultaneous knockdown of the sNPF peptide (the equivalent of sNFPR knockdown, see *Figure 1—figure supplement 1C*) and DTKR in Orco neurons abolished the effect of starvation, leading to behavior indistinguishable from that of fed flies (*Figure 1E* and *Figure 1—figure supplement 1D*). Thus, these two modulation systems are both required to bring about the appetitive behavior observed in starved flies.

## Starvation modulation of glomerular activity

To identify a circuit-level mechanism for modulation of food search behavior, we next examined whether corresponding changes in glomerular activity could be detected. We used two-photon microscopy to monitor odor-evoked activity in the second order projection neurons (PNs) that receive input from ORNs (*Wang et al., 2003*). Flies bearing *GH146-LexA* and *LexAop-GCaMP* transgenes allow the imaging of PN dendritic calcium responses in specific glomeruli to cider vinegar (*Figure 2A*). Given that the PN response in DM1 to low odor concentrations is sensitized by starvation (*Root et al., 2011*), we compared the response to higher odor concentrations in fed and starved flies. Strikingly, starvation suppresses olfactory sensitivity of DM5 (*Figure 2B*) a glomerulus that mediates aversion (*Semmelhack and Wang, 2009*). Furthermore, DM5 is the only glomerulus recruited by vinegar that is modulated by starvation at high odor intensity (see *Figure 2—source data 1* for a more complete characterization of glomerular responses to vinegar in fed and starved flies). Testing a range of concentrations, we found that DM5 became activated at 20% saturated vapor (SV pressure) in fed flies, but not until 80% SV in starved flies (*Figure 2C*). At the concentrations when DM5 is suppressed by starvation, DM1 responses saturate and do not exhibit further modulation by starvation (*Figure 2C*). Thus, modulation of DM5 olfactory sensitivity by starvation increases the activation threshold of a glomerulus that mediates aversion.

In light of our RNAi results (*Figure 1D*), we hypothesized that the starvation-dependent suppression of olfactory sensitivity in DM5 is controlled by DTKR. To test this idea, we imaged PN responses to cider vinegar while knocking down DTKR in most ORNs in flies bearing the *GH146-LexA, LexAop-GCaMP, Orco-Gal4,* and *UAS-DTKR-RNAi* transgenes. Consistent with our hypothesis, knockdown of DTKR in ORNs abolished the starvation-dependent suppression in DM5 across a range of odor concentrations (*Figure 2D*). DM1 responses to 80% SV vinegar, however, remained unaffected. The lack of an effect of

**Table 1**. Differentially expressed GPCRs in the antennae of fed and starved flies

| FlyBase ID | Gene | Gene name | Count ratio | p-value | FPKM starved |
|---|---|---|---|---|---|
| A. Receptors for biogenic amines and related compounds | | | | | |
| FBgn0011582 | DopR | Dopamine receptor | 1.52 | 0.000 | 2.59 |
| FBgn0053517 | D2R | Dopamine 2-like receptor | 1.32 | 0.001 | 1.63 |
| FBgn0038980 | oa2 | Octopamine receptor 2 | 1.29 | 0.000 | 19.64 |
| FBgn0038542 | TyrR | Tyramine receptor | 1.23 | 0.020 | 0.85 |
| FBgn0004168 | 5-HT1A | Serotonin receptor 1A | 1.21 | 0.000 | 8.27 |
| FBgn0250910 | Octbeta3R | Octbeta3R | 1.20 | 0.000 | 17.00 |
| FBgn0037546 | mAChR-B | muscarinic Acetylcholine Receptor, B-type | 1.18 | 0.000 | 10.28 |
| FBgn0004514 | Oct-TyrR | Octopamine-Tyramine receptor | 1.16 | 0.021 | 2.09 |
| FBgn0087012 | 5-HT2 | Serotonin receptor 2 | 1.15 | 0.000 | 10.50 |
| FBgn0024944 | Oamb | Octopamine receptor in mushroom bodies | 1.15 | 0.000 | 37.92 |
| FBgn0000037 | mAcR-60C | muscarinic Acetylcholine receptor 60C | 1.14 | 0.000 | 6.74 |
| FBgn0035538 | DopEcR | Dopamine/Ecdysteroid receptor | 1.08 | 0.000 | 133.42 |
| FBgn0015129 | DopR2 | Dopamine receptor 2 | 1.07 | 0.044 | 4.67 |
| FBgn0038063 | Octbeta2R | Octbeta2R | 0.79 | 0.003 | 1.30 |
| B. Peptide receptors | | | | | |
| FBgn0039396 | CcapR | Cardioacceleratory peptide receptor | 2.42 | 0.017 | 0.13 |
| FBgn0004622 | Takr99D | Tachykinin-like receptor at 99D | 1.67 | 0.029 | 0.28 |
| FBgn0003255 | rk | rickets | 1.50 | 0.000 | 0.98 |
| FBgn0033579 | CG13229 | – | 1.45 | 0.002 | 1.78 |
| FBgn0053696 | CNMaR | CNMamide Receptor | 1.44 | 0.019 | 0.51 |
| FBgn0036934 | sNPF-R | short neuropeptide F receptor | 1.41 | 0.000 | 7.26 |
| FBgn0028961 | AlstR | Allatostatin receptor | 1.30 | 0.011 | 0.85 |
| FBgn0035331 | DmsR-1 | Dromyosuppressin receptor 1 | 1.26 | 0.003 | 1.59 |
| FBgn0038880 | SIFR | SIFamide receptor | 1.20 | 0.000 | 2.78 |
| FBgn0259231 | CCKLR-17D1 | CCK-like receptor at 17D1 | 1.13 | 0.000 | 68.50 |
| FBgn0025631 | moody | moody | 1.09 | 0.000 | 50.86 |
| FBgn0016650 | Fsh | Fsh-Tsh-like receptor | 1.08 | 0.021 | 7.74 |
| FBgn0085410 | TrissinR | Trissin receptor | 1.06 | 0.025 | 15.30 |
| FBgn0038874 | ETHR | ETHR | 0.94 | 0.003 | 21.31 |
| FBgn0031770 | CG13995 | – | 0.91 | 0.000 | 15.93 |
| FBgn0004841 | Takr86C | Tachykinin-like receptor at 86C | 0.91 | 0.016 | 6.86 |
| FBgn0029723 | Proc-R | Proctolin receptor | 0.89 | 0.005 | 6.22 |
| FBgn0030954 | CCKLR-17D3 | CCK-like receptor at 17D3 | 0.79 | 0.000 | 8.62 |
| FBgn0025595 | AkhR | Adipokinetic hormone receptor | 0.74 | 0.000 | 11.06 |
| FBgn0038201 | Pk1r | Pyrokinin 1 receptor | 0.67 | 0.000 | 11.58 |
| FBgn0039354 | Lgr3 | Lgr3 | 0.51 | 0.000 | 0.24 |
| FBgn0039595 | AR-2 | Allatostatin receptor 2 | 0.39 | 0.002 | 0.11 |
| C. Metabotropic glutamate receptor family | | | | | |
| FBgn0050361 | mtt | mangetout | 3.27 | 0.000 | 0.54 |
| FBgn0019985 | mGluRA | metabotropic glutamate receptor | 1.94 | 0.000 | 1.14 |
| FBgn0052447 | CG32447 | – | 1.89 | 0.000 | 4.27 |
| FBgn0031275 | GABA-B-R3 | GABA-B receptor subtype 3 | 1.27 | 0.000 | 2.44 |

*Table 1. Continued on next page*

Table 1. Continued

| FlyBase ID | Gene | Gene name | Count ratio | p-value | FPKM starved |
|---|---|---|---|---|---|
| FBgn0051760 | CG31760 | – | 1.17 | 0.000 | 6.36 |
| FBgn0051660 | pog | poor gastrulation | 1.16 | 0.000 | 34.68 |
| FBgn0260446 | GABA-B-R1 | GABA-B receptor subtype 1 | 1.16 | 0.000 | 33.18 |
| FBgn0085401 | CG34372 | – | 1.10 | 0.033 | 3.96 |
| FBgn0027575 | GABA-B-R2 | GABA-B receptor subtype 2 | 0.94 | 0.000 | 27.45 |

Each RNA sample was from the antennae of 200 female flies ($w^{1118}$;+;Orco-Gal4/+). Count ratio is the number of reads aligned to each gene between starved and satiated flies. FPKM, fragment per kilobase of exon per million mapped fragments. p-values were calculated on raw counts using the Fisher's exact test in edgeR package.

DTKR on DM1 responses may reflect a saturation of these responses at this odor concentration that would mask further increase in response amplitude upon reduction of DTKR levels (*Figure 2—figure supplement 1A,B*). Furthermore, starvation-dependent sensitization in DM1 was abolished by knockdown of sNPFR (*Figure 2D*), as previously reported (*Root et al., 2011*). Although two other glomeruli (VM2 and DM2) also exhibited changes in activity at a high odor concentration when DTKR signaling is removed (*Figure 2—figure supplement 1A*), it is known that these two glomeruli do not contribute to appetitive behavior (*Semmelhack and Wang, 2009*). We next asked whether the starvation modulation of DM5 generalizes to other odors, and found that the response of DM5 PNs to ethyl butyrate is similarly modulated by DTKR (*Figure 2—figure supplement 1C,D*).

What is the source of DTK peptide for DM5 suppression? Previous studies have shown that a population of GABAergic local interneurons (LNs) labeled by the *GH298-Gal4* line is immunoreactive for DTK (*Ignell et al., 2009*). We investigated whether these LNs are the source of DTK for DM5 modulation, by knocking down DTK expression in LNs with RNAi. We first monitored odor-evoked activity in flies that express GCaMP in PNs and DTK-RNAi in LNs or ORNs as a negative control. Imaging DM5 responses to cider vinegar, we found that DTK peptide knockdown in LNs abolished the starvation-dependent DM5 suppression, whereas expression of DTK-RNAi in ORNs did not have any effect (*Figure 3A*). Next, we asked if DTK expression in LNs is required for starvation-dependent behavior, and found expression of DTK-RNAi in LNs, but not ORNs, significantly reduces the appetitive index for food search behavior (*Figure 3B*). Together, these studies demonstrate that DTKR signaling is required for the starvation-induced suppression of DM5, which augments food search behavior.

## Specificity of sNPF and tachykinin modulation in antennal lobe glomeruli

Thus far we have shown that two parallel neuropeptide signaling systems are employed at different ends of the odor concentration spectrum to enhance appetitive behavior. The data suggest that sNPF and DTK receptors are preferentially upregulated by starvation in select glomeruli, such as DM1 and DM5. We therefore investigated whether DTK and sNPFRs coexist in the same ORN population, or whether each receptor selectively exerts greater effect on specific glomeruli. To test this, we first investigated the effect of exogenous application of either peptide on the activity of DM1 and DM5. We performed electrical stimulation of the olfactory nerve while measuring calcium activity in PNs before and after addition of synthetic sNPF or DTK. We found that exogenous sNPF increased the response of DM1 but not DM5 in starved flies (*Figure 4A*). Conversely, exogenous DTK suppressed activity in DM5 but not DM1 in starved flies (*Figure 4B*). In a previous report, we observed modulation of DM1 when DTKR levels were knocked down in most ORNs (*Ignell et al., 2009*). Thus our current observation that DTK peptide administration doesn't decrease responses in DM1 may be due to saturation of DTKR already present in Or42b by endogenous levels of the peptide; whereas upregulation of DTKR in Or85a may explain its enhanced response to the DTKR peptide in this population. It is noteworthy that other glomeruli such as DM2 and VM2 exhibit some DTK sensitivity even in the satiated state, as observed in the current and previous study (*Ignell et al., 2009*). Thus, DM1 and DM5 appear to be specifically sensitive to the addition of sNPF and DTK, respectively, in both a glomerular-specific and starvation-dependent manner.

To further investigate whether this starvation-dependent neuropeptide modulation is specific to DM1 and DM5, we imaged PN responses while knocking down neuropeptide receptor genes in their corresponding ORNs—Or42b or Or85a, respectively. Knockdown of sNPFR in Or42b neurons

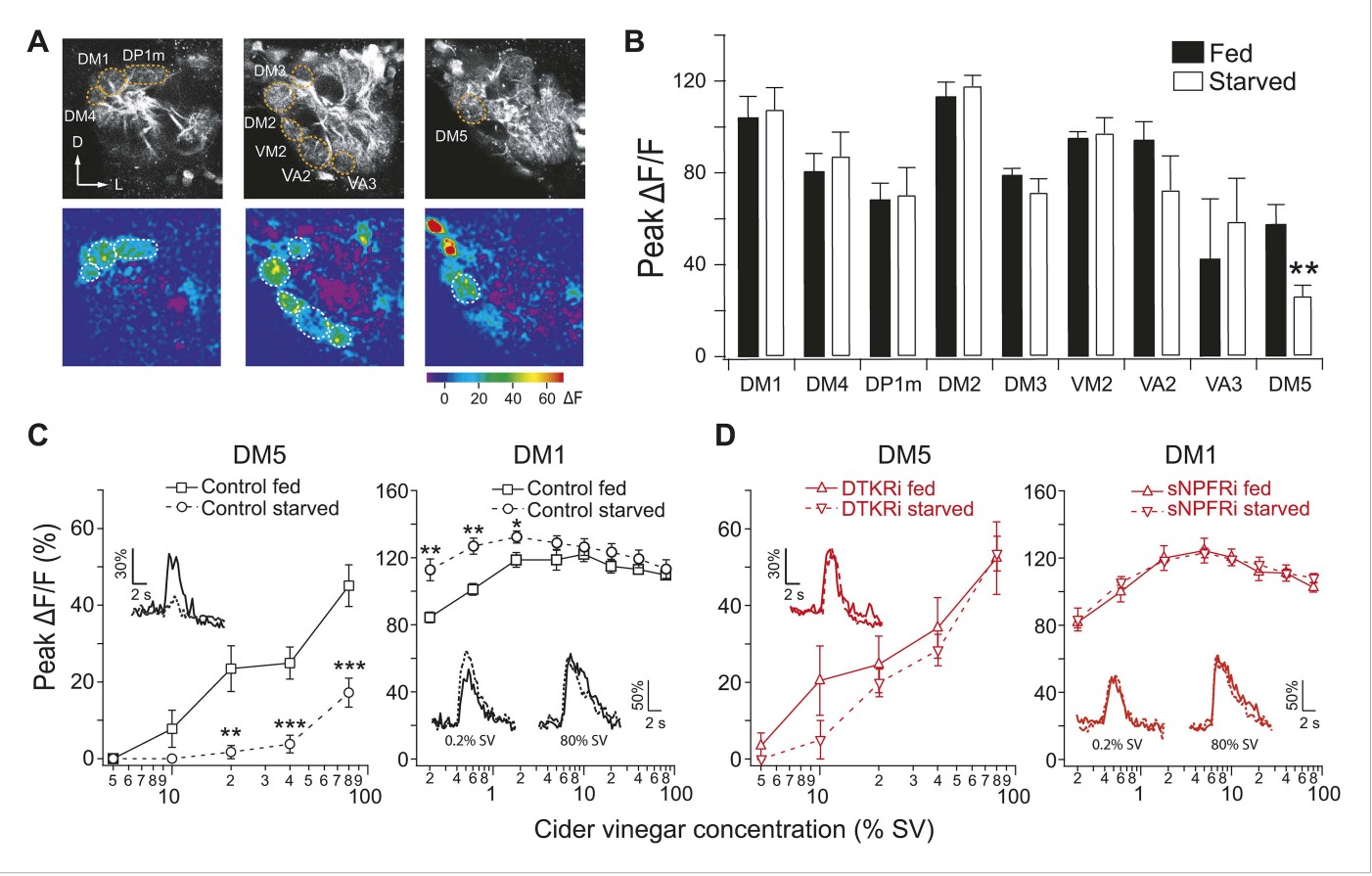

**Figure 2**. Starvation-dependent neuropeptide signaling modulates sensitivity of the DM1 and DM5 glomeruli. (**A**) Representative two photon images of projection neuron (PN) dendritic calcium responses to 80% saturated vapor (SV) of apple cider vinegar in starved DTKR knockdown flies. Grey-scale images show the glomerular map on three optical planes whereas the pseudocolored images show the change of fluorescence. (**B**) Peak ΔF/F in glomeruli that are activated by 80% SV cider vinegar in fed and starved flies. (**C**, **D**) Responses in the DM1 and DM5 glomeruli at varying concentrations in fed and starved control flies (**C**) and in flies that express *DTKR-RNAi* or *sNPFR-RNAi* in ORNs labeled by *Orco-Gal4* (**D**). Calcium signals for PN responses were imaged using *GH146-LexA, LexAop-GCaMP* flies in addition to the indicated transgenes. n = 5–7 for each. Error bars show s.e.m. *p < 0.05, **p < 0.01, ***p < 0.001; Student's *t*-test comparing between starved and fed responses.

The following source data and figure supplement are available for figure 2:

**Source data 1**. Glomerular responses to vinegar in fed and starved flies.

**Figure supplement 1**. PN responses to vinegar in flies with DTKR knockdown.

abolished starvation-dependent sensitization in DM1 (*Figure 4C*) and reduced the appetitive index to the same extent as when the RNAi construct was expressed in most ORNs (*Figure 4D*). Thus, reducing sNPFR dependent modulation of Or42b activity blocks the effects of starvation in enhancing both neuronal sensitivity and behavioral attraction. According to our working model, behavioral attraction at higher odor concentrations of vinegar is the sum of the opposing effects of Or42b and Or85a. We propose that removing the sNPFR modulation of Or42b does not reduce behavioral attraction to fed levels because the weight of Or42b increases when Or85a is inhibited. This working model is supported by our observation that net behavioral attraction is not completely abolished by genetic knockdown of sNPFR. Knocking down DTKR had no effect in the same neurons for which we observed phenotypes with sNPFR-RNAi. Although DTKR could in theory be absent from this population, we prefer the hypothesis Or42b responses are saturated by the 5% vinegar that we used for these behavioral experiments.

Likewise, knockdown of DTKR in Or85a neurons abolished starvation-dependent suppression of DM5 and reduced food finding to the same extent as when the RNAi construct was expressed in

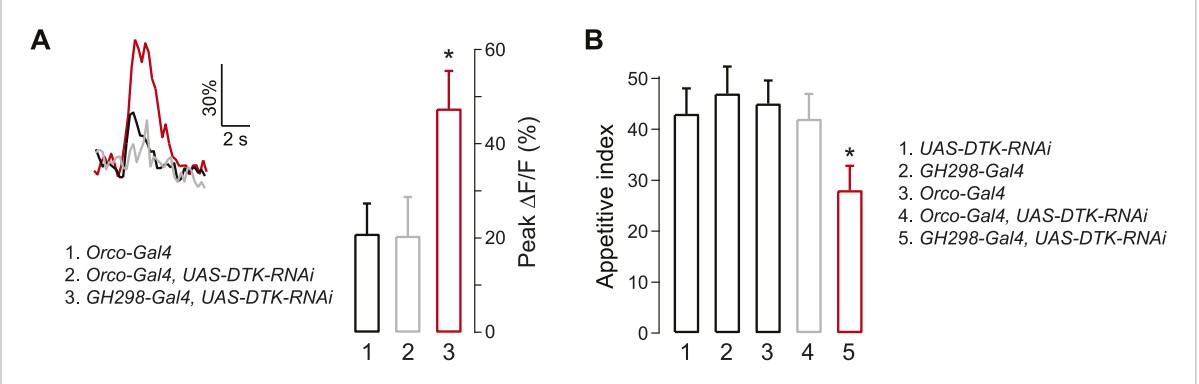

**Figure 3**. Tachykinin released by antennal lobe local interneurons (LNs) is necessary for starvation-dependent suppression of DM5 glomerular activity. (**A**) Representative traces showing ΔF/F in the DM5 glomerulus in flies that have *UAS-DTK-RNAi* in ORNs (*Orco-Gal4*) or in LNs (*GH298-Gal4*). Bar graph depicts peak ΔF/F for each indicated genotype. Calcium signals for PN responses were imaged using *GH146-LexA, LexAop-GCaMP* flies in addition to the indicated transgenes. n = 5 for each condition. (**B**) The appetitive index of DTK knockdown flies in response to 5% cider vinegar. n = 87–101 for each condition. For imaging experiments, Error bars show s.e.m. *p < 0.05, Student's t-test (**A**) and z-test for proportions (**B**) comparing *GH298-Gal4, UAS-DTK-RNAi* group to control groups or *Orco-Gal4, UAS-DTK-RNAi* group.

most ORNs. Thus, reducing DTKR dependent modulation of Or85a activity blocks the effects of starvation on both neuronal sensitivity and behavioral attraction. Expression of sNPFR-RNAi in the same neurons had no effect. Taken together, these findings indicate that sNPF and DTK modulate distinct ORNs in opposite directions, in what appears to be a push–pull optimization strategy to increase the attractive valence of an odor.

Based on these results, we predicted that artificial enhancement of DM1 and DM5 should shift the appetitive behavior. In particular, sensitization of DM1 in fed flies should mimic the effect of starvation, while sensitization of DM5 in starved flies should mimic the effect of satiety. To test this hypothesis, we ectopically expressed the bacterially derived sodium channel (NaChBac), which makes neurons hyperexcitable (*Nitabach et al., 2006*). Targeted expression of NaChBac in Or42b neurons increased the olfactory sensitivity of DM1 in fed flies and resulted in a marked increase in appetitive index (*Figure 4E*). Likewise, expression of NaChBac in Or85a neurons increased activity of DM5 in starved flies and was accompanied by a significant decrease in appetitive index (*Figure 4F*). Interestingly, activation of Or85a drives down levels of behavioral attraction, but does not trigger behavioral repulsion, presumably because Or42b that is still present provides competing inputs. Thus, by increasing activity in DM1 or DM5, we were able to directly influence foraging behavior in opposite directions in a manner that mimics behavior appropriate for the corresponding metabolic state.

What is the metabolic sensor for starvation to suppress DM5's response to food odor? Previous work has implicated insulin signaling in mediating differences between rover and sitter, naturally occurring polymorphisms in the foraging gene that lead to dramatic differences in feeding behaviors (*de Belle et al., 1989*; *Kent et al., 2009*). We also recently reported that insulin negatively regulates sNPFR gene expression in DM1 ORNs (*Root et al., 2011*). To determine whether insulin also functions as the upstream metabolic cue regulating DTKR signaling in the Or85a/DM5 neurons, we investigated the effect of insulin signaling on the olfactory sensitivity of DM5 and appetitive behavior. We first blocked insulin signaling with wortmannin, an inhibitor of PI3K (*Weinkove et al., 1999*; *Root et al., 2011*). Flies fed with sugar and wortmannin exhibited a suppressed DM5 response that was accompanied by an increased appetitive index (*Figure 5A*), thereby mimicking the starved state. Moreover, this suppression occurs through increased DTKR signaling, because targeted knockdown of DTKR in ORNs blocked the effect of wortmannin. We next asked whether constitutive activation of InR prevents starvation-like physiology and behavior. Indeed, ectopic expression of a constitutively active InR (InR-CA) in Or85a neurons of starved flies led to increased DM5 activity and decreased appetitive behavior (*Figure 5B*), thereby mimicking satiety. Thus, insulin signaling in Or85a neurons gates the expression of DTKR to modulate DM5 activity and appetitive behavior.

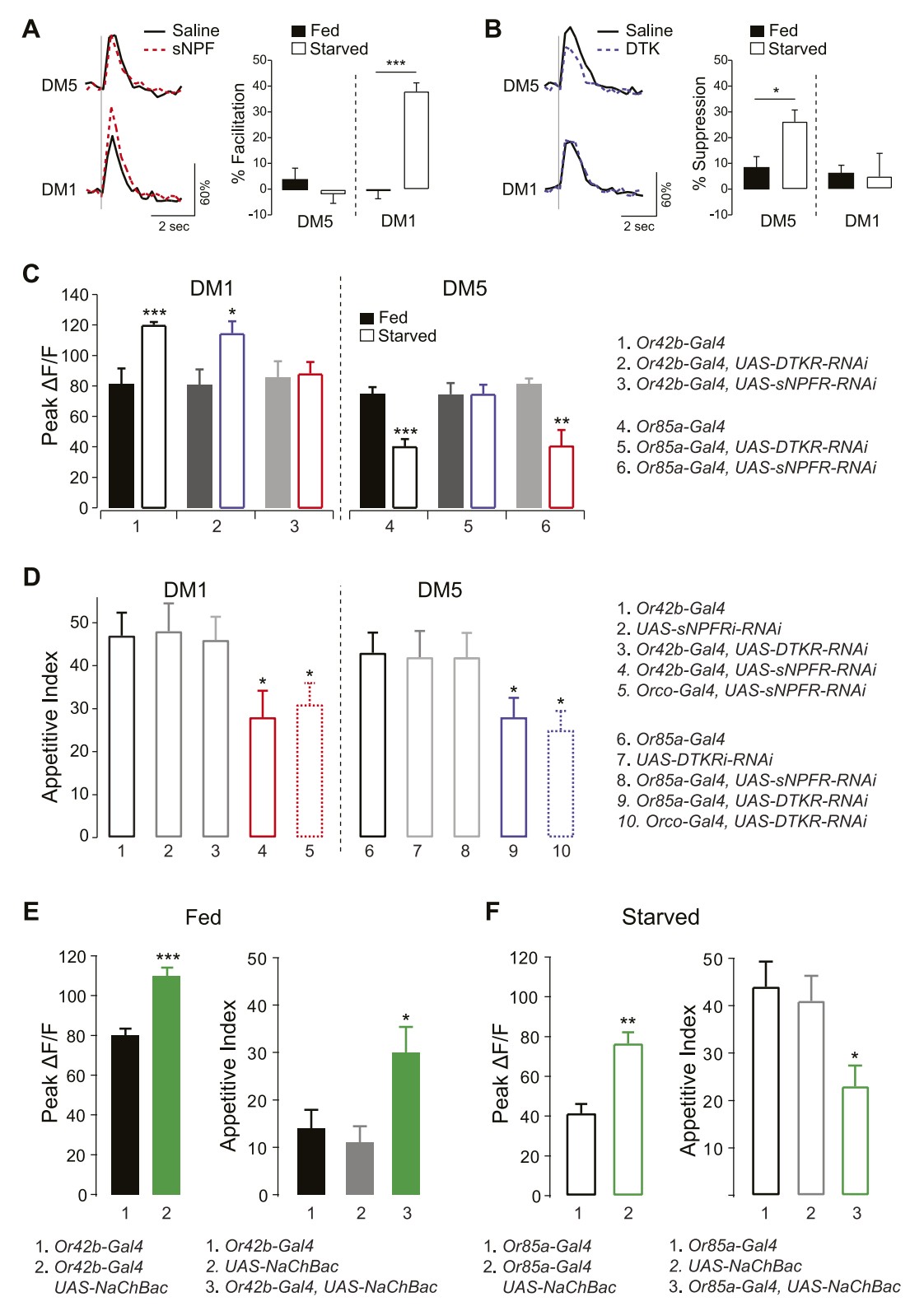

**Figure 4**. sNPF and DTK modulatory mechanisms target different sensory neurons. (**A**, **B**) Representative traces of calcium activity (left) in the DM1 and DM5 PNs in response to olfactory nerve stimulation before and after bath application of sNPF (**A**) or DTK (**B**) synthetic peptides. The percent facilitation and suppression (right) are measured as the percent change in peak ΔF/F after peptide addition. (**C**) Peak ΔF/F responses in the DM1 PNs (left) and the DM5 PNs (right) to cider vinegar with glomerulus specific neuropeptide receptor knockdown. (**D**) The appetitive index of starved flies with glomerulus

*Figure 4. continued on next page*

*Figure 4. Continued*

specific neuropeptide receptor knockdown. (**E**, **F**) Expression of NaChBac sensitizes DM1 and DM5 (left), which alters appetitive behavior in opposite directions (right). For imaging experiments (n = 5–6), 0.2% and 80% SV cider vinegar was used to stimulate the DM1 and DM5 PNs, respectively. For the behavioral experiments (n = 53–111), 5% cider vinegar was used. Error bars show s.e.m. *p < 0.05, **p < 0.01, ***p < 0.001; *t*-test for the imaging results, comparing between starved and fed responses; z-test for the behavioral results comparing knockdown groups to Gal4 or UAS control groups.

## Discussion

Here we demonstrate that shifts in the internal metabolic state of an animal lead to dramatic functional changes in its olfactory circuit and behaviors. Starved flies exhibit enhanced odor sensitivity in ORNs that mediate behavioral attraction and decreased sensitivity in ORNs that mediate behavioral aversion. This functional remodeling of the olfactory map is mediated by parallel neuromodulatory systems that act in opposing directions on olfactory attraction and aversion. In our earlier study, we showed that sNPFR signaling increases sensitivity in Or42b ORNs and thus enhances behavioral attraction (*Root et al., 2011*). In our current study, however, we show that sNPFR signaling does not account for all changes induced by starvation in behavioral responses to a wider range of odor concentrations. Second, we show that starvation leads to a decreased sensitivity in the Or85a ORNs, an odorant channel that mediates behavioral aversion (*Semmelhack and Wang, 2009*). Third, we show that DTKR signaling mediates the reduced sensitivity in the Or85a ORNs and partly accounts for enhanced behavioral attraction to high concentrations of vinegar. Fourth, we show eliminating DTKR and sNPFR signaling pathways together fully reverses the effect of starvation on behavioral attraction across all odor concentrations tested. Finally we show evidence suggesting that the same global insulin signal regulating sNPFR expression may also regulate DTKR expression.

In the wild, rotten fruits early in the fermentation process are more attractive to *Drosophila* than fresh or highly fermented fruits (*Chakir et al., 1996*; *Castrezana and Markow, 2001*). In the laboratory, well fed flies display very little attraction to apple cider vinegar (*Root et al., 2011*). Low levels of vinegar are indicative of fresh fruit of limited nutritional value. Expanding odor sensitivity to lower concentrations of potential food odors may encourage flies to accept food sources of lower value. High odor concentrations typically accompany late stages of fermentation and are often aversive or uninteresting to flies. We show that starved flies are attracted to high concentrations of vinegar partly due to neuromodulatory mechanisms that enhance sensitivity in Or42b ORNs, an attractive odor channel, and partly through neuromodulatory mechanisms that reduce sensitivity in Or85a ORNs, an aversive odor channel. In our working model, behavioral attraction to higher odor concentrations of vinegar is the sum of the opposing effects of Or42b and Or85a (*Figure 6B,C*). When flies face starvation, the balance of these inputs shifts to favor Or42b over Or85a inputs, as mediated by selective upregulation of sNPFR and DTKR in these ORNs, respectively (*Figure 6D,E*). These processes could serve to encourage flies to risk ingestion of potentially toxic foods when under nutritional stress.

Given the broad array of glomeruli that can respond to odors such as vinegar (*Figure 2—source data 1*), it may be surprising that the modulation of only two glomeruli is sufficient to significantly impact fly behavioral attraction. Whether these findings extend to a broad array of food associated odors and whether additional glomeruli are modulated by these neuromodulatory systems remain to be determined. In this context, we note that a recent correlational analysis predicts DM5 activity is highly correlated with behavioral attraction (*Knaden et al., 2012*). However, this prediction has not been confirmed by direct testing of the DM5 glomerulus in behavioral experiments and is contradicted by more recent findings (*Gao et al., 2015*), as well as the data in this paper. Thus our findings suggest that in starved flies the concentration range over which vinegar odor is attractive expands in both directions, with the acute need for caloric intake apparently outweighing considerations of food quality or risk (*Figure 6C*).

This study highlights the importance of neuromodulators in shaping local neural circuit activity to accommodate the internal physiological state of an organism. The often unique expression patterns of specific GPCRs in sensory systems highlights the flexibility conferred by this evolutionarily ancient mechanism to translate neuroendocrine signals into local shifts in neuronal

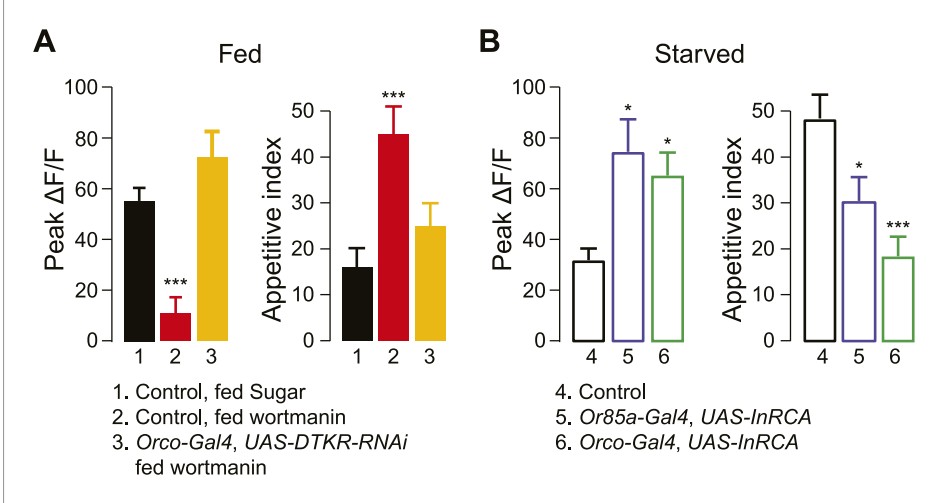

**Figure 5.** Insulin controls DTKR signaling. (**A**) Peak ΔF/F responses of the DM5 glomerulus to cider vinegar (left) and the appetitive index of flies (right) that were fed overnight with 4% sucrose alone or sucrose with the PI3K blocker, wortmannin. DTKRi flies contained the *DTKR-RNAi* in *Orco-Gal4*. (**B**) Peak ΔF/F responses of the DM5 glomerulus to cider vinegar (left) and the appetitive index of starved flies (right) that expressed constitutively active insulin receptor (InR-CA) in the *Orco* ORNs or selectively in Or85a neurons. For imaging experiments, PN responses to 80% SV cider vinegar were measured using *GH146-LexA, LexAop-GCaMP* flies. n = 5–13 for each condition. For behavior experiments, fly responses to 5% apple cider vinegar were measured. n = 67–91 for each condition. Error bars show s.e.m. *p < 0.05, **p < 0.01, ***p < 0.001; Student's *t*-test (imaging results) and z-test for proportions (behavioral results) comparing wortmannin-fed group to sugar-fed group (**A**), and comparing the InR-CA groups to the control counterpart (**B**).

excitability and network properties that ultimately lead to adaptive behaviors. sNPF shares structural and functional similarities with its vertebrate homolog, NPY (*Hewes and Taghert, 2001*; *Lee et al., 2004*). Both neuropeptides show roles in controlling food intake and feeding behaviors in insects and vertebrates. Interestingly, NPY is also expressed in the vertebrate olfactory bulb (*Hansel et al., 2001*; *Mathieu et al., 2002*; *Mousley et al., 2006*) and is thus positioned to shape olfactory processing during shifts in appetitive states as well. sNPF's broad expression pattern in the fly brain (*Nassel et al., 2008*) supports the possibility it is widely used to orchestrate changes across many different neuropils to shape appetitive behaviors. Indeed, sNPF and NPF, another NPY homolog in *Drosophila*, have been shown in the fly gustatory system to control sweet and bitter taste sensitivity, respectively, in parallel but opposing directions (*Inagaki et al., 2014*). The similar changes manifested by nutritional stress in both the olfactory and gustatory systems suggests complex networks of neuromodulators may shape sensory processing of aversive and attractive inputs differentially throughout the brain in a hunger state.

DTK and DTKR share homology with substance P and its receptor NK1, respectively (*Li et al., 1991*). Interestingly, they seem to share roles in shaping the processing of stressful or negative sensory cues in both flies and mammals. For example, in rodents, emotional stressors cause long-lasting release of substance P to activate NK1 in the amygdala to generate anxiety-related behavior (*Ebner et al., 2004*). In *Drosophila*, DTK signaling has also been shown to be critical for aggressive behaviors among male flies (*Asahina et al., 2014*). In previous work, we showed *Drosophila* tachykinin mediates presynaptic inhibition in ORNs and detected expression in the LNs (*Ignell et al., 2009*). In this current study, we map the locus of DTK's effects on behavioral responses to vinegar to the Or85a/DM5 ORNs using behavior and functional imaging. We also confirm that the source of the peptide is indeed the LNs as previous anatomical data had suggested (*Ignell et al., 2009*). Thus, tachykinin's role in modulating stressful sensory inputs appears to extend to a glomerulus hardwired to behavioral aversion in the olfactory system.

Our results here resonate with discoveries in the gustatory system (*Inagaki et al., 2014*) and show that starvation changes the perception of both attractive and aversive sensory inputs beginning at the

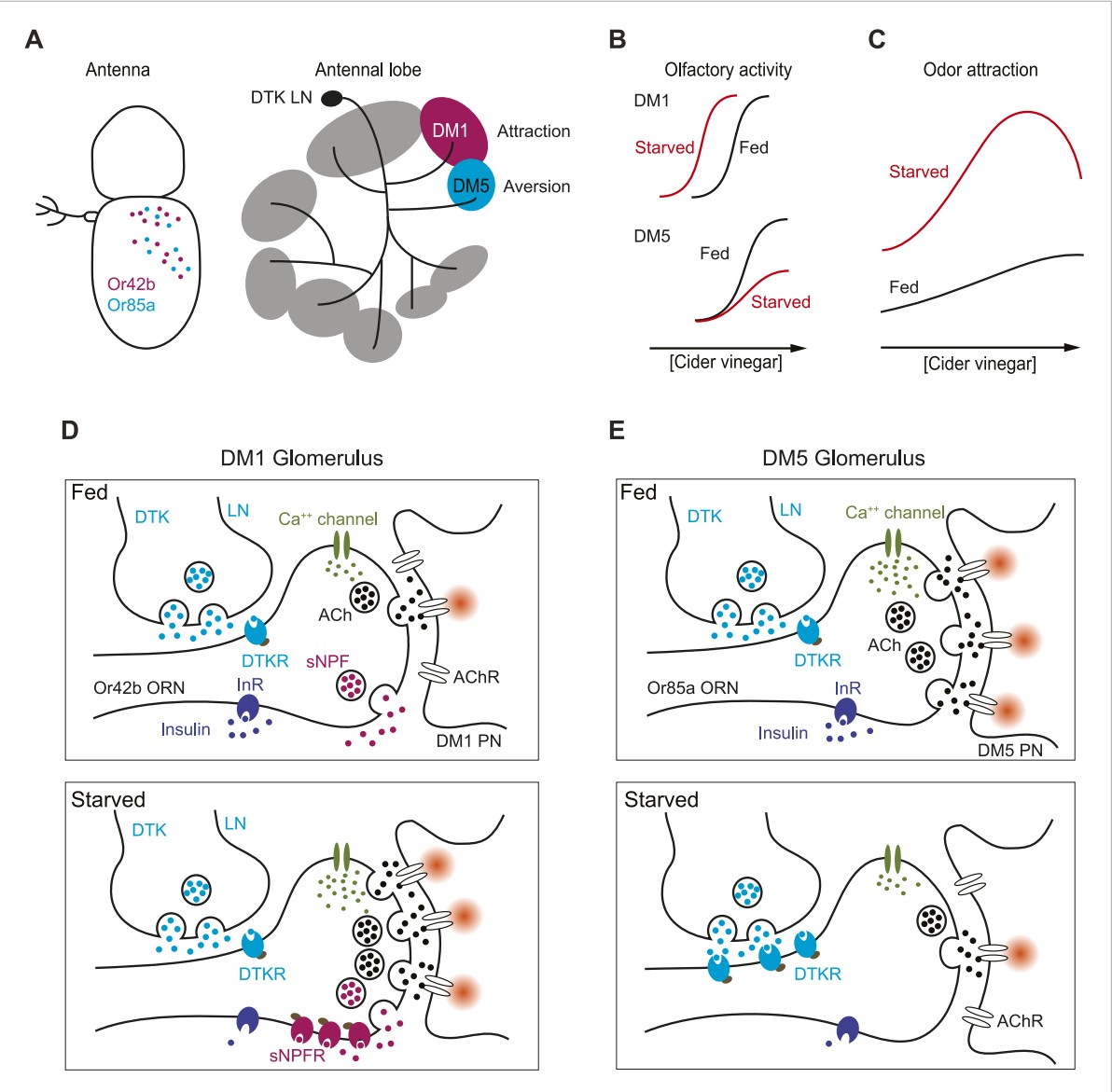

**Figure 6**. How starvation changes early olfactory processing. (**A**) A schematic diagram depicting anatomical locations for Or42b and Or85a ORNs in the fly antenna as well as their corresponding glomeruli, DM1 and DM5, respectively, in the antennal lobe. LNs release DTK peptide broadly throughout the antennal lobe. (**B**) A model for how starvation state fine-tunes ORN sensitivity via the actions of neuromodulation of Or42b/DM1 and Or85a/DM5. In the starvation state, sNPF sensitizes the DM1 glomerular responses through additive gain modulation. Tachykinin suppresses DM5 glomerular responses through a divisive gain modulation. (**C**) The concerted effect of these two neuromodulatory systems increases behavioral attraction and expands the concentration range over which attraction to vinegar manifests. (**D**) In the DM1 glomerulus, both DTK and sNPF are available and released from the LNs and ORNs, respectively, in both the fed and starved states. DTKR is also present in these terminals in the fed state. Upon starvation, loss of insulin signaling leads to selective upregulation of sNPFR expression in the Or42b ORNs, which leads to their presynaptic facilitation. (**E**) In the DM5 glomerulus, both DTK and sNPF are available and released from the LNs and ORNs, respectively, in both the fed and starved states. Upon starvation, loss of insulin signaling leads to upregulation of DTKR expression in the Or85a ORNs, leading to their presynaptic inhibition.

peripheral nervous system. Through the use of parallel neuromodulatory systems, the internal state of the organism functionally reconfigures early olfactory processing to optimize its detection of nutrients at the risk of ignoring potentially toxic food resources. It is certainly likely that neuromodulatory systems also impact and reconfigure central circuits in appetitive contexts. Thus, it will be of great interest to understand the contributions of peripheral and central circuits towards modifying appetitive behaviors.

## Materials and methods

### Transgenic flies

All *Gal4-* and *UAS-* control flies were crossed to $w^{1118}$ fly strain. The following fly stocks were used: *Orco-Gal4* (*Kreher et al., 2005*); *Or42b-Gal4*, *Or85a-Gal4* (II) (*Fishilevich and Vosshall, 2005*); *GH146-LexA* (*Lai et al., 2008*), *LexAOp-GCaMP* (*Root et al., 2008*); *UAS-sNPFR-RNAi* (*Lee et al., 2008*); *UAS-DTKR-RNAi* and *GH298-Gal4* (*Ignell et al., 2009*); *UAS-DTK-RNAi* (*Winther et al., 2006*); *UAS-InR-CA*, *Or42b-Gal4* (III) and *Or85a-Gal4* (III) (Bloomington stock center #8263, #9972 and #24461); *UAS-NaChBac* (*Nitabach et al., 2006*).

### Behavior assay

Single-fly assay was used to measure the latency of food finding as previously described (*Root et al., 2011*; *Zaninovich et al., 2013*). Female flies that were 2–5 days old and presumed non-virgin were used for all experiments. Single flies were introduced into chambers that were 60 mm in diameter and 6 mm in height. The chamber was illuminated by 660 nm LEDs. Flies were tracked at 2 Hz with custom software written in Labview (V.8.5, National Instruments, Austin, TX), and analysis was performed with Igor Pro (V.6, Wavemetrics, Inc., Portland, OR) using a custom macro (*Root et al., 2011*; *Zaninovich et al., 2013*). Apple cider vinegar was diluted in 1% low melting temperature agarose. 5 µl of cider vinegar solution was placed in the center of the chamber for all experiments. A fly was counted as having found the food when it spends 5 s or longer within a 5 mm radius of the center. The elapsed time before an individual fly reached the odor target was also recorded. All control flies were crossed with $w^{1118}$ flies. Flies were starved with water for 16–24 hr prior to experiments.

### RNA-seq

About 50 flies ($w^{1118}$;+;*Orco-Gal4/+*) of both sexes were kept in each vial for 3 days. Female flies were then transferred to a new food vial (control fed flies) or a vial with a Kimwipe saturated by water (starved flies). 12 hr later, antennae were collected from these female flies. Dissection was performed in the morning at the same time to minimize circadian difference. Antennae from 200 flies were collected for each condition and total RNA was extracted using Trizol (Invitrogen, Carlsbad, CA). Libraries were prepared using Illumina's mRNA sequencing kit and further purified using AMPure XP beads (Agencourt). Sequencing was performed at UCSD's BIOGEM facility on an Illumina GA2 sequencer. For each of the biological conditions, over 80 million 36 bp reads were generated from two lanes. Reads were aligned to the *Drosophila* genome (dm3 assembly) using TopHat (*Trapnell et al., 2009*), allowing up to three mismatches with the reference sequence. Transcripts were then assembled against FlyBase (release 5.39) gene annotations and their abundances were calculated using Cufflinks (*Trapnell et al., 2010*). In total, over 136 million reads were mapped to protein-coding genes. For differential expression analysis, raw gene counts were generated using HTSeq (*Anders, 2010*) software and then normalized for the difference in sequencing depth between the two conditions. Probability values were calculated on raw counts using the Fisher's exact test as computed by the edgeR package (*Robinson et al., 2010*) (R software environment).

### Two-photon calcium imaging

GCaMP imaging was performed as previously described (*Wang et al., 2003*; *Root et al., 2008*). In odor experiments, a constant airflow of 1 l/min was applied to the antennae via a pipe of 12 mm diameter. Odor onset was controlled by mixing a defined percentage of carrier air with air redirected through odor bottles as previously described (*Root et al., 2008*; *Semmelhack and Wang, 2009*). Nerve stimulation was performed with a glass suction electrode and an S48 stimulator (Grass, Warwick, RI) as previously described (*Wang et al., 2003*; *Root et al., 2008*). Stimulation was 1 ms in duration, 10 V in amplitude, and 16 pulses (*Figure 4A*) and 45 pulses (*Figure 4B*) at 100 Hz. Starved flies were starved with water for 16–24 hr.

### Pharmacology

sNPF peptide, AQRSPSLRLRF-NH$_2$, 98% purity (Celtek Peptides, Franklin, TN) and DTK peptide, APTSSFIGMR-NH$_2$, 98% purity (Bio Basic Inc., Markham, Ontario, Canada) were each dissolved in saline to a final concentration of 10 µM. Wortmannin (LC Laboratories, Woburn, MA) was dissolved in DMSO at stock concentrations of 10 mM. Flies were fed overnight with 200 µl of 4% sucrose solution, or plus 25 nM wortmannin.

# Additional information

## Funding

| Funder | Grant reference | Author |
| --- | --- | --- |
| National Institute on Deafness and Other Communication Disorders (NIDCD) | R01DK092640 | Jing W Wang |
| National Science Foundation (NSF) | 0920668 | Jing W Wang |
| National Institute on Deafness and Other Communication Disorders (NIDCD) | R01DC009597 | Jing W Wang |
| National Institute of General Medical Sciences (NIGMS) | R01GM050545 | Steven A Wasserman |
| National Institute on Deafness and Other Communication Disorders (NIDCD) | 1F31DC009511 | Cory M Root |

The funders had no role in study design, data collection and interpretation, or the decision to submit the work for publication.

## Author contributions

KIK, CMR, Conception and design, Acquisition of data, Analysis and interpretation of data, Drafting or revising the article; SAL, Acquisition of data, Analysis and interpretation of data, Drafting or revising the article; OAZ, AKS, Acquisition of data, Drafting or revising the article; SAW, JWW, Conception and design, Drafting or revising the article; SMK, Analysis and interpretation of data, Drafting or revising the article

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
