## [Decision Letter]

Thank you for submitting your work entitled “Starvation promotes concerted modulation of appetitive olfactory behavior via parallel neuromodulatory circuits” for peer review at *eLife*. Your submission has been favorably evaluated by K VijayRaghavan (Senior editor), Mani Ramaswami (Reviewing editor), and two other reviewers.

The reviewers have discussed the reviews with one another and the Reviewing editor has drafted this decision to help you prepare a revised submission.

Summary:

The authors show that feeding and starvation influence insulin signaling, which acts through two spatially and molecularly distinct neuromodulatory pathways to differentially influence innately attractive and repulsive olfactory glomeruli in the insect antennal lobe.

Starvation causes increased appetitive behavior, measured elegantly as the speed and efficiency with which flies converge on a drop of apple cider vinegar (ACV, a probable food source). In addition, starvation changed ACV (or EB) induced responses in attractive and repulsive glomeruli, previously identified by [51]. Responses in the repulsive DM5 glomerulus are decreased, at the same time as responses in the attractive DM1 glomerulus are increased.

In another previous paper (48) the Wang lab showed that starvation-induced modulation of DM1 at a relatively low AMV concentration was mediated by the sNPF pathway. In this work, the authors elegantly show that sNPF signaling underlies the increase in appetitive tracking at low but not at higher vinegar concentrations.

What is the mechanism to explain increased appetitive tracking at high odor concentrations in starved flies? The authors show that starvation-induced tachykinin signaling onto DM5 causes both the increase response to vinegar at higher concentrations. Tachykinin required for this is released by the GH298 class of LNs, previously shown to express and release DTK onto OSNs. Here this is also shown to be regulated by starvation and insulin signaling, and to have a specific inhibitory effect on the DM5 glomerulus in starved flies.

The effect requires receptors for sNPF and DTK to function in cognate sensory neurons. Thus, together the authors’ observations, nicely documented, delineate two parallel neuromodulatory pathways that independently on OSNs mediating attractive and repulsive signals.

There are, however, some concerns and recommendations.

Essential revisions:

1) The imaging data presented focus exclusively on the few glomeruli that this and prior work show to be relevant to attraction and repulsion. This is not unreasonable. However, practitioners in the field would be interested to see how other glomerular responses are affected by hunger, and satiety, and to know whether and how DTK or sNPF neuromodulatory activity is involved in these changes. Is there no effect in some glomeruli? Is it variable and difficult to interpret in others? These data, which must already be available, should be presented in tabular form or as supplementary data. For completeness, if necessary, the table can also include previously published data (with due citation). A brief discussion of the table will also be useful to include.

2) In this context, it is also useful to discuss the use of behavioral observations to make inferences regarding “attraction” and “repulsion.” In Semmelheck et al., 2009, the Wang lab used a 4-field chamber assay, in which flies show obvious repulsion from EB or vinegar under conditions where DM5 is activated and attraction when it is not (and when DM1 is active). This is broadly consistent with findings in a recent PLoS One paper from Liqun Luo's groups. In contrast, in the assay used in this manuscript, even when DM5 is activated (or DM1 is silenced), the flies show a net attraction to the food source. However, this attraction is greater when DM1 is active and DM5 is silent (and vice versa). Interestingly, observations using this assay are broadly consistent with data from Bill Hansson's extended group, who inferred that DM5 is attractive.

Both for a balanced presentation and to illuminate the field, the authors should both cite the relevant work from the Jena group and attempt to rationalize these apparently conflicting observations into a single model.

3) Previous work on DTKR in the fly olfactory system (68) suggested a broader role of this receptor in mediating pre-synaptic inhibition in the antennal lobe (including DM1). The current work deals with this problem directly, and makes a strong case that starvation dependent DTK modulation of the ACV response occurs through pre-synaptic inhibition of DM5 by DTK. The manuscript should integrate Ignell et al., and discuss or account for the possibility that different odor maps may be modulated more extensively by DTK.

4) The RNAseq experiment is well designed and allowed the authors to quickly zoom in onto DTKR. On the other hand the long list of GPCRs with expression changes is not discussed in any detail. Can these data be better integrated with the rest of the paper?

5) The model at the end is really too simple and almost a cartoon. The authors should include a thoughtful and detailed model that contains all pathways and circuits including hypothesized molecular signals, their sources, effects and their anatomical sites of action.

---

## [Author Response]

*1) The imaging data presented focus exclusively on the few glomeruli that this and prior work show to be relevant to attraction and repulsion. This is not unreasonable. However, practitioners in the field would be interested to see how other glomerular responses are affected by hunger, and satiety, and to know whether and how DTK or sNPF neuromodulatory activity is involved in these changes. Is there no effect in some glomeruli? Is it variable and difficult to interpret in others? These data, which must already be available, should be presented in tabular form or as supplementary data. For completeness, if necessary, the table* can *also include previously published data (with due citation). A brief discussion of the table will also be useful to include*.

We included a table ([Supplementary-material SD1-data]) that summarizes our current (Figure 2—figure supplement 1) and published imaging data (48) for glomerular responses to vinegar in fed and starved flies. We have also included in this table our data for glomerular responses to vinegar in starved flies carrying transgenes for either *sNPF-RNAi* or *DTKR-RNAi* driven by *Orco-Gal4*.

As summarized in both this and the [48] paper, glomeruli that exhibit significant increases in their responses to vinegar include DM1, DM4 and DM2. Glomeruli that show significant decreases in their responses to vinegar after starvation include VM2, VA3 and DM5. Reducing sNPFR expression in ORNs reversed the effects of starvation on glomerular responses to vinegar in DM1, DM4, DM2 and VM2. Reducing DTKR expression in ORNs led to significant increases in glomerular responses to vinegar in VM2 and DM5; unexpectedly, it also led to significant decreases in DM2.

In the Results section, we have included a sentence “A more complete characterization of glomerular responses to vinegar in fed and starved flies is included in [Supplementary-material SD1-data]”.

*2) In this context, it is also useful to discuss the use of behavioral observations to make inferences regarding “attraction” and “repulsion.” In Semmelheck et al., 2009, the Wang lab used a 4-field chamber assay, in which flies show obvious repulsion from EB or vinegar under conditions where DM5 is activated and attraction when it is not (and when DM1 is active). This is broadly consistent with findings in a recent PLoS One paper from Liqun Luo's groups. In contrast, in the assay used in this manuscript, even when DM5 is activated (or DM1 is silenced), the flies show a net attraction to the food source. However, this attraction is greater when DM1 is active and DM5 is silent (and vice versa). Interestingly, observations using this assay are broadly consistent with data from Bill Hansson's extended group, who inferred that DM5 is attractive*.

*Both for a balanced presentation and to illuminate the field, the authors should both cite the relevant work from the Jena group and attempt to rationalize these apparently conflicting observations into a single model*.

First, in this manuscript, we do not present any data where Or42b/DM1 activity is completely abolished. Instead, we merely block the modulation of Or42b/DM1 activity with a sNPFR-RNAi transgene and demonstrate that this blocks the effect of starvation on the sensitivity of Or42b ORNs and on behavioral attraction. This is an important distinction, because DM1 remains responsive to vinegar even when sNPFR levels are reduced as demonstrated by our imaging data (Figure 2). Thus, according to our working model, because higher odor concentrations of vinegar recruit Or85a/DM5, behavioral attraction is the sum of the opposing effects of Or42b and Or85a. Removing sNPFR modulation of Or42b does not reduce behavioral attraction to the same level of fed flies because the weight of Or42b increases when Or85a activity is reduced by DTKR. This working model is supported by our observations that net behavioral attraction is not completely abolished by genetic knockdown of sNPFR. We clarify this issue in our text with the inclusion of the following statements in the Results section:

“Thus, reducing sNPFR dependent modulation of Or42b activity blocks the effects of starvation in enhancing both neuronal sensitivity and behavioral attraction. According to our working model, behavioral attraction at higher odor concentrations of vinegar is the sum of the opposing effects of Or42b and Or85a. We propose that removing the sNPFR modulation of Or42b does not reduce behavioral attraction to fed levels because the weight of Or42b increases when Or85a is inhibited. This working model is supported by our observation that net behavioral attraction is not completely abolished by genetic knockdown of sNPFR”.

“Interestingly, activation of Or85a drives down levels of behavioral attraction, but does not trigger behavioral repulsion, presumably because Or42b that is still present provides competing inputs”.

Second, we also respectfully disagree with the suggestion that our results conflict with the conclusion about DM5’s role in behavioral aversion in the Semmelhack (2009) and Gao (2015) papers. Both papers directly demonstrate causality between DM5 activity and behavioral aversion using behavioral analysis and precise genetic manipulation of DM5 activity and show behavioral aversion is observed only when competing inputs from all (or nearly all) other odorant channels are removed. In our current study, a net behavioral attraction to vinegar is observed, even when DM5 activity is enhanced with NaChBac, because DM1 activity remains intact and contributes to the net behavioral response.

Thus the fly’s behavioral response reflects a sum of inputs from DM1 and DM5. These data from both the Wang and Luo groups contradict inferences made by the Hansson group that DM5 glomerular activity is largely predictive of odor attraction. It is difficult to reconcile the Hansson group’s conclusions with our own or the Luo group without knowing more about the conditions under which DM5 would be attractive. The Jena group’s apparent contradiction in the behavioral role of DM5 needs to move beyond correlation and be substantiated with manipulation of DM5 directly in a behavioral analysis. We acknowledge their contributions however, in the Discussion (bottom paragraph) section with the following text:

“Given the broad array of glomeruli that can respond to odors such as vinegar ([Supplementary-material SD1-data]), it may be surprising that the modulation of only two glomeruli is sufficient to significantly impact fly behavioral attraction. Whether these findings extend to a broad array of food associated odors and whether additional glomeruli are modulated by these neuromodulatory systems remain to be determined. In this context, we note that a recent correlational analysis predicts DM5 activity is highly correlated with behavioral attraction (29). However, this prediction has not been confirmed by direct testing of the DM5 glomerulus in behavioral experiments and is contradicted by more recent findings (17), as well as the data in this paper”.

*3) Previous work on DTKR in the fly olfactory system (*[68]*) suggested a broader role of this receptor in mediating pre-synaptic inhibition in the antennal lobe (including DM1). The current work deals with this problem directly, and makes a strong case that starvation dependent DTK modulation of the ACV response occurs through pre-synaptic inhibition of DM5 by DTK. The manuscript should integrate Ignell et al., and discuss or account for the possibility that different odor maps may be modulated more extensively by DTK*.

We are grateful for this particular comment on the Ignell paper. We realized that we may have overreached in our interpretation of our data and in the statement that DTKR may not be expressed in Or42b. In the Ignell paper, we showed that Or42b sensitivity to odors increases when DTKR is knocked down with an RNAi transgene. In our current manuscript, however, we do not see any effect on Or42b sensitivity to vinegar after a DTKR-RNAi transgene is selectively targeted to this population of ORNs. The simplest and most plausible explanation for this discrepancy is likely due to the saturation of Or42b ORN responses in starved flies to the concentrations of vinegar (0.2% SV) used in these experiments (Figure 4). Indeed, our imaging data demonstrates that Or42b responses in starved flies plateau and are fairly close to maxed out at all vinegar concentrations tested, and thus may explain why further increases with DTKR knockdown were not observed (Figure 2).

In the DTKR peptide experiments (Figure 4), we do not see additional suppression of Or42b responses, possibly, because all the DTKR receptors present on Or42b have been saturated with endogenous DTKR levels. We speculate that DTKR is upregulated in only select glomeruli such as Or85a during a starved state. Therefore, we have revised the manuscript to state that our results are consistent with a view that DTKR mediated presynaptic inhibition is widespread in different glomeruli, but starvation increases DTKR expression in specific glomeruli such as DM5.

In Results section for Figure 4:

“In a previous report, we observed modulation of DM1 when DTKR levels were knocked down in most ORNs (68). Thus our current observation that DTK peptide administration doesn’t decrease responses in DM1 may be due to saturation of DTKR already present in Or42b by endogenous levels of the peptide; whereas upregulation of DTKR in Or85a may explain its enhanced response to the DTKR peptide in this population”.

In Results section for Figure 4:

“Knocking down DTKR had no effect in the same neurons for which we observed phenotypes with sNPFR-RNAi. Although DTKR could in theory be absent from this population, we prefer the hypothesis Or42b responses are saturated by the 5% vinegar that we used for these behavioral experiments”.

4) The RNAseq experiment is well designed and allowed the authors to quickly zoom in onto DTKR. On the other hand the long list of GPCRs with expression changes is not discussed in any detail. Can these data be better integrated with the rest of the paper?

We have incorporated additional text commenting on select GPCRS identified in our RNAseq experiment in the Results section:

“This analysis identified 34 GPCRs that have higher expression in the antennae of starved flies […] thus making these receptor classes highly plausible targets for internal state modulation.”

“We also identified 11 GPCRs that exhibited reduced expression in the antennae of starved flies (Table 1). This group included a few GPCRs described as having roles in feeding behaviors or energy storage. One example is as the adipokinetic hormone receptor, which regulates lipid and carbohydrate storage (7). Another is the pyrokinin 1 receptor, which is activated by hugin, a neuropeptide that suppresses feeding (40).”

*5) The model at the end is really too simple and almost a cartoon. The authors should include a thoughtful and detailed model that contains all pathways and circuits including hypothesized molecular signals, their sources, effects and their anatomical sites of action*.

In our revised manuscript, we have included a more detailed schematic (Figure 6) illustrating the molecular signals and antennal lobe circuits involved in our study of starvation mediated changes in early olfactory processing.